# Thermally Reduced Graphene Oxide/Thermoplastic Polyurethane Nanocomposites: Mechanical and Barrier Properties

**DOI:** 10.3390/polym13010085

**Published:** 2020-12-28

**Authors:** Santiago Maldonado-Magnere, Mehrdad Yazdani-Pedram, Héctor Aguilar-Bolados, Raul Quijada

**Affiliations:** 1Facultad de Ciencias Químicas y Farmacéuticas, Universidad de Chile, Olivos 1007, 8380544 Santiago, Chile; santiago.maldonado@usach.cl (S.M.-M.); myazdani@ciq.uchile.cl (M.Y.-P.); 2Facultad de Ciencias Físicas y Matemáticas, Universidad de Chile, Beauchef 851, 8370456 Santiago, Chile; haguilar@ciq.uchile.cl

**Keywords:** reduced graphene oxide, thermoplastic elastomers, mechanical and barrier properties

## Abstract

This work consists of studying the influence of two thermally reduced graphene oxides (TRGOs), containing oxygen levels of 15.8% and 8.9%, as fillers on the barrier properties of thermoplastic polyurethane (TPU) nanocomposites prepared by melt-mixing processes. The oxygen contents of the TRGOs were obtained by carrying out the thermal reduction of graphene oxide (GO) at 600 °C and 1000 °C, respectively. The presence and contents of oxygen in the TRGO samples were determined by XPS and their structural differences were determined by using X-ray diffraction analysis and Raman spectroscopy. In spite of the decrease of the elongation at break of the nanocomposites, the Young modulus was increased by up to 320% with the addition of TRGO. The barrier properties of the nanocomposites were enhanced as was evidenced by the decrease of the permeability to oxygen, which reached levels as low as −46.1%.

## 1. Introduction

Carbon-based polymer nanocomposites are versatile materials for a number of purposes such as environmental and energy applications [1,2,3]. This is mainly due to the unique properties of carbon-based nanomaterials (such as graphene-based materials) which exhibit high electrical and thermal conductivities, excellent mechanical properties, and light weight [4]. For instance, the use of reduced graphene nanomaterials as fillers in a polymer matrix imparts an enhancement of electrical and thermal conductivity as well as an improvement of the mechanical properties of the resulting nanocomposites [5,6,7]. Electrically conductive polymer nanocomposites are attractive because they can be used for different purposes, such as electromagnetic interference shielding or as electrostatic discharge materials and thermoelectric generators [8,9,10]. In spite of the diversity of polymer matrices used for the preparation of carbon-based nanocomposites, thermoplastic elastomers stand out due to their unique mechanical properties [11]. In this regard, thermoplastic elastomers are polymers that present good processability and have comparable elastomeric properties to those of vulcanizable thermoset rubbers [12].

Among thermoplastic elastomers, thermoplastic polyurethanes (TPUs) stand out because of their massive potential for use in multiple applications [13,14]. Overall, alternating segments of polyol and isocyanate form the TPUs backbone. Ether links of polyol segments impart flexibility to the polymer backbone, while the aromatic rings of isocyanate segments provide rigidity to the backbone. Therefore, the polyol hetero-chains are referred to as soft segments, and those of isocyanate as hard segments [15]. Besides, the interaction between the hydrogen atoms of amine groups and carbonyl oxygen leads to a hydrogen-bonded structure with long-range order. Consequently, these interactions strongly influence the mechanical, electrical, and thermal properties of polyurethanes [16]. 

As mentioned, the addition of fillers to polymer matrices allows the enhancement of mechanical and electrical properties. The effectiveness of this strategy depends on several factors such as degree of the filler dispersion in the polymer matrix, affinity between the filler and the polymer matrix, and filler shape and its aspect ratio, among others [12]. The affinity between the filler and the matrix stems from the nature of the functional groups present on the filler surface, which can promote or inhibit the interactions of the filler with the polymer [17]. Polyurethane elastomers are polar in nature, thus polar fillers will present affinity with these polymers. The polar character of the fillers is imparted by the moieties such as oxygenated functional groups i.e., alcohol, ether, or ketones. Carbon-based nanomaterials are widely used as fillers. Thermally reduced graphene oxide (TRGO) prepared at a high temperature is a graphene material with a low content of polar functional groups [18]. TRGO can be produced by a bottom-up process, by using graphite as starting material. This process consists of two stages. Firstly the oxidation of graphite is carried out by using the methods reported by Brodie, Hummers, or Tour, and then the graphene oxide is thermally reduced to obtain TRGO through a thermal process carried out at temperature above 400 °C under inert atmosphere and without using reducing agents such as hydrazine [18,19,20,21,22]. In order to favor the restoring of the long-range π-conjugated system, the thermal reduction is generally carried out at a high temperature [23]. When the thermal reduction is carried out at temperatures as high as 1000 °C, the majority of the oxygenated functional groups are eliminated [24]. Hence, to carry out the reduction process at a lower temperature leads to a higher oxygen content in the TRGO. In this regard, the content of oxygenated functional groups in TRGO can be modulated by controlling the temperature of the reduction process. In this work, the influence of the oxygen content of TRGO on the mechanical, electrical, and barrier properties of polyurethane elastomer is studied.

## 2. Materials and Methods 

### 2.1. Materials

Natural graphite powder (>50 µm), hydrochloric acid (32%), sulfuric acid (98.0%), sodium nitrate (NaNO_3_), potassium permanganate (99%), and hydrogen peroxide (3%) were supplied by Merck (Kenilworth, New Jersey, USA). The TPU used was Elastollan 1185 A, supplied by Basf (Ludwigshafen, Germany). Indura (Santiago, Chile) supplied the high purity oxygen (99.999%).

### 2.2. Preparation of Graphene Oxide and Thermally Reduced Graphene Oxide

Graphene oxide was prepared by adding 10 g of graphite, 5 g of NaNO_3_, and 200 mL of sulfuric acid (98.0%) to a round-bottom flask and the mixture was stirred for a duration of 30 min. Then, the reaction temperature was brought to 0 °C and 30 g of potassium permanganate was carefully added and left to react for 4 h. Once the reaction time was concluded, the reaction temperature was increased to 25 °C and left whilst the mixture was stirred for 30 min. Then the reaction mixture was slowly added to 500 mL of distilled water, followed by the addition of 450 mL of 3% H_2_O_2_. Then, the reaction product was filtered, and the resulting solid was washed with 200 mL of concentrated hydrochloric acid. Finally, the solid product was dried under vacuum at 110 °C for 5 h. 

In order to prepare the TRGO, 0.5 g of dry graphene oxide was deposited into a quartz reactor. The reactor was purged by using nitrogen gas and then sealed. The reactor was then introduced into a vertical tube furnace at 600 °C or 1000 °C. After 40 s, the reactor was removed from the furnace. The resulting products were designated as TRGO_600_ and TRGO_1000_, where the subscript indicates the reduction temperature.

### 2.3. Preparation of Nanocomposites

The nanocomposites were prepared by using a Brabender Lab Station Torque Rheometer Plasti-Corder (Duisburg, Germany), using a W 30/50 EHT measuring mixer. The blade type was roller. The mixing temperature, mixing time, and rotor speed were 210 °C, 5 min, and 90 rmp, respectively. The filler content in the polymer matrix was varied between 0 and 7 wt.%. Films of nanocomposites were prepared by using a HP hydraulic press with heated plates (HP industries, Buenos Aires, Argentina). The dimensions (length × height × width) of the two types of prepared nanocomposites films were 110 mm × 110 mm × 1.1 mm and 110 mm × 110 mm × 0.15 mm. The first type of film was used to obtain the specimens for tensile characterization and the second type was used for determining the permeability of composites.

### 2.4. Characterization of Carbon-Based Nanomaterials

The chemical compositions of graphite and graphene materials were determined by using a Perkin Elmer elemental analyzer MCHNSO/ 2400. The mass used for each sample was 2.0 mg. In addition, the Brunauer-Emmett-Teller (BET) surface areas of these carbon-based materials were determined using nitrogen gas in a Nova Station A analyzer, Quantachrome Instrument (Boynton Beach, FL, USA). The mass of each sample for this analysis was 3.0 mg. The X-ray photoelectron spectroscopy of the carbon-based materials was carried out at 400 W with an emission angle of 70°, using a Perkin Elmer XPS-Auger spectrometer, model PHI 1257 (Waltham, MA, USA), which included an ultra-high vacuum chamber, a hemispheric electron energy analyzer, and an X-ray source with Kα radiation unfiltered from an Al anode (*hν* = 1486.6 eV). The C1s and O1s photoelectrical lines were studied to analyze the contribution of the functional groups. Besides, the Raman spectra of the carbon-based materials were recorded from 0 to 4000 cm^−1^ using a Renishaw Invia Raman microscope (Gloucestershire, United Kingdom) equipped with a laser wavelength of 514.5 nm and resolution of 0.02 cm^−1^. The dry powder was placed on a quartz sample holder to carry out the measurement. Each Raman spectrum was normalized with respect to its highest band intensity. The crystallite sizes (*L_a_*) of different graphitic materials were calculated using a reported procedure [25]. 

X-ray diffraction analyses of the graphitic materials were performed by using a Bruker diffractometer model D8 Advance (Billerica, MA, USA). The incident angle (*2θ*) was varied between 2° and 80° and the scan rate was 0.02 °/s with a Cu Kα radiation source, wavelength λ = 0.154 nm and power supply of 40 kV and 40 mA. The interlayer distance (*d*_00l_), the crystallite size (*D_00l_*), and the number of stacked layers (*N_L_*) of different materials was estimated using a reported procedure [25].

### 2.5. Characterization of the Nanocomposites

Tensile tests (ASTM D-412) on nanocomposite samples were carried out by using an Instron Universal Testing System model 3382, (Norwood, MA, USA) at an elongation speed of 50 mm/min. The distance between the jaws and the specimen thickness were 40 mm and 1 mm, respectively. Each sample test was quintuplicated. Besides, the fracture zone of each of these specimens was analyzed using an Inspect F50 field-emission scanning electron microscope, FEI (Hillsboro, OR, USA). The barrier properties to the oxygen of nanocomposites were analyzed using a manometric gas permeability tester, Lyssy L–100–5000 (Devens, MA, USA). 

## 3. Results and Discussion

### 3.1. Structure and Properties of Thermally Reduced Graphene Oxide

The surface area and elemental analysis of graphene oxide (GO) and different TRGO samples are shown in Table 1. GO presents the highest oxygen content, indicating the success of the oxidation process. As expected, the oxygen content in TRGO decreases as the temperature of the reduction process is increased. The TRGO obtained at 1000 °C presents the lowest oxygen content. The increase of the oxygen content because of the reduction process indicates the functionalization of a graphene layer with oxygen moieties, where this has an impact on the structural features of the material. Figure 1 presents the normalized X-ray diffraction analysis of different samples in a waterfall plot. The oxidation of graphite induces the distance increases between the individual layers of graphene. As a result, the peak associated to the (002) plane is shifted to a lower angle. As seen in Figure 1, the reduction processes of different TRGOs allows the partial restoration of the (002) plane, which is associated with the recovery of the interlayer distance similar to that observed for graphite [26]. However, the width of the diffraction peaks associated to the (002) plane indicate the loss of crystallinity. Undoubtedly, this is related to the occurrence of the exfoliation process experienced by the stacked graphene oxide layers.

The angle of the diffraction plane, interplanar distance (*d*), crystal size (*D*), and the number of stacked graphene layers (*N_L_*) estimated for the different samples in Table 2 are shown. It is observed that graphite presents a peak at 26.3°, which corresponds to the (002) plane. This is associated with the distance between the graphene layers. Graphite presents an interlayer distance of 0.338 nm, a crystal size of 18.61 nm, and 56 stacked layers. Because of the oxidation process, the interlayer distance observed in GO is 0.723 nm, which is higher than that observed in graphite. The decrease in both the crystal size and the number of stacked layers is also observed. This indicates that the oxidation process not only favors the functionalization of the graphene layer with oxygen moieties which increases the interlayer distance, but it also reduces the crystallinity and favors the partial delamination of graphite. The temperature of the reduction process had an impact on the interlayer distance of the reduced graphene oxide layers. TRGO_600_ presents a distance of 0.355 nm, a crystal size of 3.71 nm, and 11 stacked layers, while TRGO_1000_ presents a peak associated to the (002) plane and the interlayer distance is similar to that of graphite, but the number of stacked layers is 13, which is lower than in graphite. 

Figure 2 presents the Raman spectra of different samples. It is observed that graphite presents typical *D, G, D’,* and *2D* bands, which are observed at 1353 cm^−1^, 1578 cm^−1^, 1618 cm^−1^, and 2708 cm^−1^, respectively (Table 2). The *D* band in GO is intense, which indicates a decrease of edge defects and the presence of oxygenated functional groups, while the *G* band is shifted to a lower frequency [25,27]. The increase in the intensity of the *G* band width also suggests the occurrence of a structural change, such as those related to the chemical nature and crystallinity of the material. The increases of the intensity of the *D’* band is also associated to the edge defect and functionalization. The TRGO shows a similar spectrum, but the increase in the intensity of the *D* and *D’* bands suggests an increase of edge defects in the graphitic structure, caused by the partial reduction of the graphene oxide. This characteristic is more visible in the TRGO_1000_ spectrum, where the intensity of the *D* band is similar to that of the G band, while the *D’* band is stronger than that observed in TRGO_600_. The inter-defects distance (*La*) was determined by using an equation previously reported [25]. It is possible to observe that the inter-defects distance of TRGO_1000_ is the lowest. This indicates that the thermal reduction process produces important graphenic structural damage.

The GO, TRGO_600_, and TRGO_1000_ samples were analyzed by using X-ray photoelectron spectroscopy (XPS). This technique allows one to determine the chemical species and to quantify the atomic percentages of carbon and oxygen present on the surface of the samples, by analyzing the C1s and O1s photoelectric lines, respectively. Table 3 details the values obtained for the atomic percentages of carbon and oxygen in the samples. It is possible to observe that GO is composed of 77.33% carbon and 22.67% oxygen. The oxygen content in GO is higher than that reported for graphite, which indicates the effectiveness of the oxidation process. TRGO_600_ and TRGO_1000_ presented oxygen contents of 10.15% and 5.35%, respectively, which implies that the temperature of the reduction process enables the control of oxygen content in the structure of TRGOs. Figure 3 shows the high resolution XPS spectrum of the C1s region of the GO and TRGOs. In the GO sample the main contributions corresponded to C–O, which can correspond to ether, alcohol, or epoxy functional groups. The contribution of different oxygen groups decreases because of the reduction process, but in both the TRGO_600_ and TRGO_1000_ samples, the oxygen contributions are similar. The O1s region provides interesting information because the contribution of C–O in the TRGO_600_ is almost twice that which is observed for the TRGO_1000_ (Figure 4).

### 3.2. Structure and Properties of Thermally Reduced Graphene Oxide

The influence on the mechanical properties of the content and type of filler used in the polymer matrices was studied through tensile tests, which provided the stress–strain curves of the samples. The results are presented in Table 4, where it is possible to observe the Young modulus, tensile strength, and elongation at break of different samples. The addition of TRGO as low as 1 wt.% to the polymer matrix produces a significant increase of the Young modulus. This is attributed to the effect of the rigid fillers, which promote the increase in the stiffness of the composites. However, the increase of the Young modulus also is associated with the decrease of the tensile strength and elongation at break of the samples. The filler dispersion and the affinity of the filler with the polymer matrix are some aspects which can be associated with the decrease of these properties. Figure 5 presents the SEM images of the fracture section of tensile test specimens of TPUs and TPU composites containing 3 wt.% of TRGO_600_ and TRGO_1000_. It is possible to observe that the graphenic fillers promote changes in the fracture characteristics, which can be associated with the state of the dispersion of the filler and the difference of the stiffness between the matrix and filler. The morphology change observed for the samples indicates that the fracture of samples is favored by the presence of the filler. The presence of the filler probably hinders the creeping of the polymer chains and eventually inhibits the interactions among polymer polar moieties. A relevant result is associated with the elongation at break presented by those composites containing TRGO_600_: the elongation at break and the tensile strength is superior to those presented by composites containing TRGO_1000_. This can be attributed to the higher content of oxygen moieties in TRGO_600_, which would promote the increase of the interaction between the filler and the polymer matrix. Taking into consideration the strong decrease of the elongation at break showed by the nanocomposites and their drastic morphology changes, it is possible to infer that the filler disrupts the hydrogen bonding between polyurethane segments [28]. This likely affects the hydrogen-bonded structure with long-range order and consequently the elastomeric features imparted by the rigid and soft domains are partially lost [16].

Table 5 presents the relationship between the oxygen permeability and the content of the TRGO_600_ and TRGO_1000_ in TPUs. It is possible to observe that 1 wt.% and 3 wt.% of TRGO_600_ increases the permeability of oxygen to 7.1% and 26.6%, respectively. However, the TRGO_600_ content of 5 wt.% and 6 wt.% produces a decrease in permeability of 16.2% and 35.7%, respectively. This indicates that with a high content of TRGO_600_ the barrier properties are enhanced due to the presence of filler, which promotes the formation of tortuous paths that hinder oxygen diffusion through the composite film. On the other hand, all of the composites containing TRGO_1000_ presented a drastic decrease in permeability, reaching levels as low as −46.1%. The difference in the permeability of composites containing TRGO_600_ and TRGO_1000_ can be understood by noting the difference in the content of oxygen functional groups in each graphitic structure. TRGO_600_ possesses a higher content of oxygen moieties, which not only can interact with the TPU matrix, but also are available to interact with oxygen gas, increasing oxygen solubility in films. Conversely, TRGO_1000_ has a lower content of oxygenated functional groups, and therefore, it has an effect mainly on the generation of tortuous paths which inhibit oxygen diffusion through the film [29].

## 4. Conclusions

The oxygen content of TRGO can be controlled by varying the temperature of the reduction process. Under an inert atmosphere, a higher temperature of the reduction process results in obtaining TRGO with lower oxygen content. Although an increase of 320% of the Young modulus was observed for a nanocomposite containing 7% of TRGO_1000_, this resulted in a drastic decrease of the elongation at break. This was attributed to the possible disruption of the hydrogen-bonded structure. In addition, a high content of oxygenated functional groups in the TRGO_600_ not only favors its interaction with the TPU matrix, but also its interaction with oxygen molecules, which increases its solubility in the nanocomposite. By contrast, TRGO_1000_ with a lower content of oxygenated functional groups imparts a nonpolar character to the nanocomposite, inhibiting the diffusion of oxygen molecules through the composite film.

## Figures and Tables

**Figure 1 polymers-13-00085-f001:**
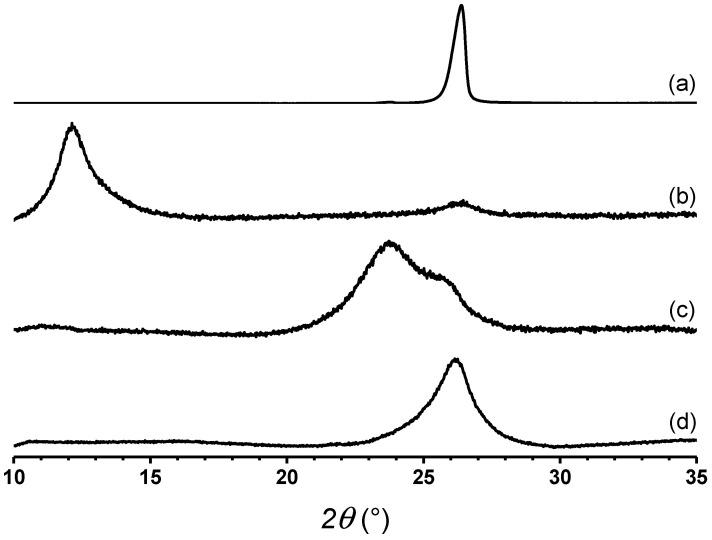
Normalized X-ray diffraction patterns of graphite (**a**), GO (**b**), TRGO_600_ (**c**), and TRGO_1000_ (**d**).

**Figure 2 polymers-13-00085-f002:**
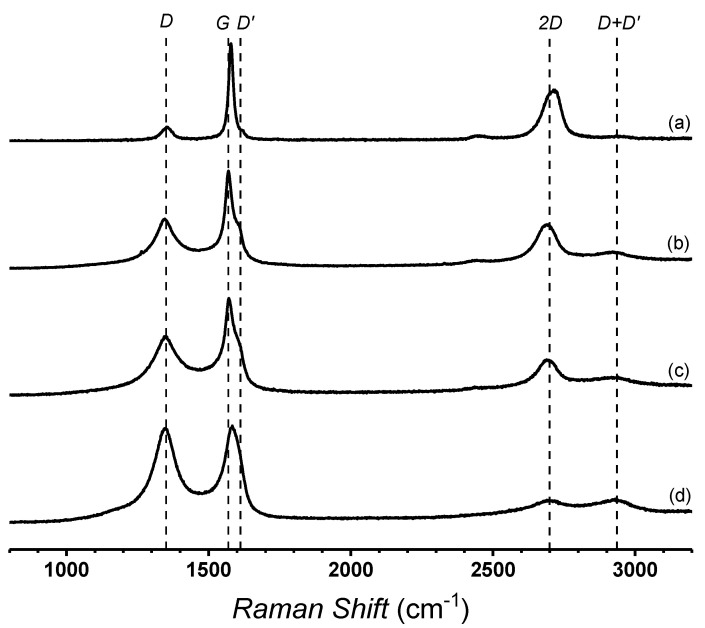
Raman spectra of graphite (**a**), GO (**b**), TRGO_600_ (**c**), and TRGO_1000_ (**d**).

**Figure 3 polymers-13-00085-f003:**
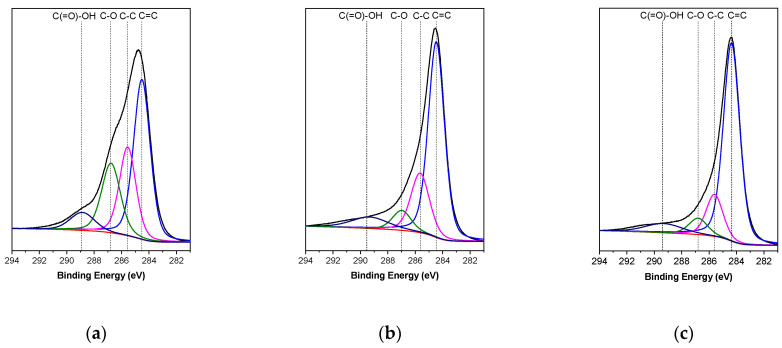
Contributions of functional groups of C1s photoelectric lines obtained from X-ray photoelectron spectroscopy (XPS) for GO (**a**), TRGO_600_ (**b**), and TRGO_1000_ (**c**).

**Figure 4 polymers-13-00085-f004:**
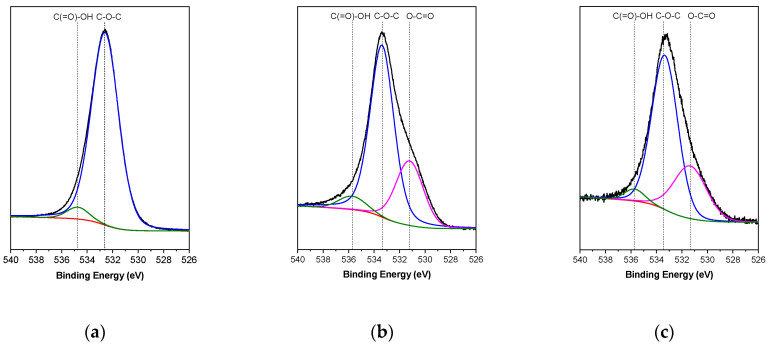
Contributions of functional groups of O1s photoelectric lines obtained from X-ray photoelectron spectroscopy (XPS) for GO (**a**), TRGO_600_ (**b**), and TRGO_1000_ (**c**).

**Figure 5 polymers-13-00085-f005:**
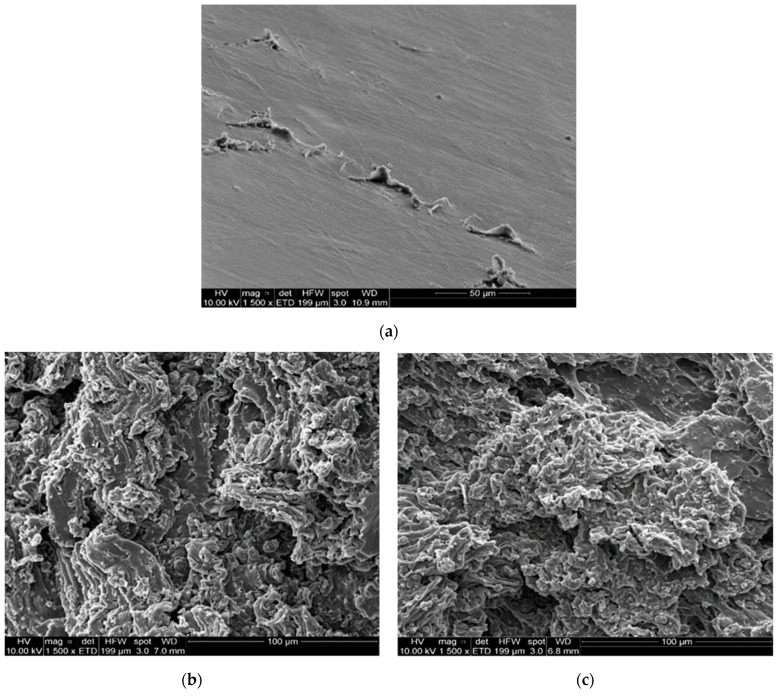
SEM images of TPU (**a**) and TPU-based nanocomposites containing 5 wt.% of TRGO_600_ (**b**) and 5 wt.% of TRGO_1000_ (**c**).

**Table 1 polymers-13-00085-t001:** BET surface area and elemental compositions of different carbon-based nanomaterials.

Sample	Surface Area (m^2^/g)	Elemental Analysis			
		C (wt.%)	H (wt.%)	O (wt.%)	C/O Ratio
**GO**	69	54.35	1.87	43.78	1.24
**TRGO_600_**	304	83.84	0.32	15.84	5.32
**TRGO_1000_**	266	90.75	0.38	8.87	10.38

**Table 2 polymers-13-00085-t002:** Structural features of graphite, GO, TRGO_600_, and TRGO_1000_ determined by X-ray diffraction and Raman spectroscopy.

Sample	X-ray Diffraction				Raman Spectroscopy					
	2*θ* (°)	*d_00l_* (nm)	*D_00l_* (nm)	*N_L_*	*D*	*G*	*D’*	*2D*	*I_D_/I_G_*	*La* *(nm)*
**Graphite**	26.3	0.338	18.61	56	1353	1578	1618	2708	0.132	128
**GO**	12.2	0.723	10.90	16	1350	1570	1608	2687	0.501	34
**TRGO_600_**	25.1	0.355	3.472	11	1352	1574	1608	2668	0.631	27
**TRGO_1000_**	26.0	0.342	4.100	13	1346	1583	1607	2681	1.006	17

**Table 3 polymers-13-00085-t003:** Signal, binding energy, assignment, area percentage, and atomic percentage of GO, TRGO_600_, and TRGO_1000_ obtained from X-ray photoelectron spectroscopy.

Sample	Signal	Binding Energy (eV)	Assignment	Area (%)	Atomic Percentage(%)
**GO**	C_1s_	284.52	C=C (sp^2^)	35.20	77.33
		285.56	C—C (sp^3^)	19.40	
		286.77	C—O	17.16	
		288.86	C(=O)—OH	5.57	
	O_1s_	532.60	C(=O)—OH	21.63	22.67
		534.73	C—O—C	1.04	
**TRGO_600_**	C_1s_	284.46	C=C (sp^2^)	55.95	89.85
		285.65	C—C (sp^3^)	19.68	
		286.99	C—O	6.57	
		289.32	C(=O)—OH	7.65	
	O_1s_	531.22	O—C=O	2.73	10.15
		533.38	C—O—C	6.89	
		535.72	C(=O)—OH	0.53	
**TRGO_1000_**	C_1s_	284.46	C=C (sp^2^)	66.97	94.83
		285.65	C—C (sp^3^)	14.61	
		286.99	C—O	6.31	
		289.32	C(=O)—OH	6.94	
	O_1s_	531.22	O—C=O	1.49	5.17
		533.38	C—O—C	3.49	
		535.72	C(=O)—OH	0.19	

**Table 4 polymers-13-00085-t004:** Mechanical properties of TPU and TPU/TRGO nanocomposites.

Content of Filler in TPU	Young Modulus (MPa)	Tensile Strength (MPa)	Elongation at Break (%)
**0 wt.%**	7.0 ± 0.6	47.6 ± 5.1	1235 ± 95
**TRGO_600_ 1 wt.%**	22.3 ± 0.9	4.7 ± 1.0	64 ± 20
**TRGO_600_ 3 wt.%**	14.6 ± 0.1	5.7 ± 0.4	235 ± 30
**TRGO_600_ 5 wt.%**	16.7 ± 0.2	12.2 ± 0.5	405 ± 22
**TRGO_600_ 7 wt.%**	20.4 ± 0.6	14.1 ± 1.4	378 ± 20
**TRGO_1000_ 1 wt.%**	18.6 ± 2.2	6.0 ± 0.6	157 ± 23
**TRGO_1000_ 3 wt.%**	16.5 ± 0.4	6.8 ± 0.5	385 ± 30
**TRGO_1000_ 5 wt.%**	16.7 ± 0.3	7.1 ± 0.5	250 ± 20
**TRGO_1000_ 7 wt.%**	22.4 ± 0.4	8.0 ± 1.8	287 ± 12

**Table 5 polymers-13-00085-t005:** Permeability of TPU and TPU-based nanocomposites.

Content of Filler in TPU	Permeability (Barrier)	Permeability Variation (%)
**0 wt.%**	1.54 ± 0.08	0
**TRGO_600_ 1 wt.%**	1.65 ± 0.12	+7.14
**TRGO_600_ 3 wt.%**	1.95 ± 0.23	+26.62
**TRGO_600_ 5 wt.%**	1.29 ± 0.05	−16.24
**TRGO_600_ 7 wt.%**	0.99 ± 0.12	−35.72
**TRGO_1000_ 1 wt.%**	0.90 ± 0.17	−41.56
**TRGO_1000_ 3 wt.%**	0.83 ± 0.12	−46.10
**TRGO_1000_ 5 wt.%**	0.89 ± 0.20	−42.20
**TRGO_1000_ 7 wt.%**	1.18 ± 0.12	−23.38

## Data Availability

Data sharing not applicable.

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
