# Peer review of "Thermally Reduced Graphene Oxide/Thermoplastic Polyurethane Nanocomposites: Mechanical and Barrier Properties"

_polymers, 2020, doi:10.3390/polym13010085_

Round 1

Reviewer 1 Report

The authors report on the preparation of “Thermally reduced graphene oxide/thermoplastic polyurethane nanocomposites: mechanical and barrier properties”

The work is quite innovative for the field of polymer nanocomposites, however several things have to be considered:

  1. Abstract: The first sentence is superfluous. Also the second sentence “The approach of this work consists of the study of the influence of”, please rewrite it since there are several “of” in series and the language usage is not appropriate.
  2. In abstract it will be nice to mention also some of the techniques performed in this work, utillised i.e to prove the reduction of GO. Moreover, please mention the method for the preparation of the nanocomposites; e.g. melt mixing, solution mixing etc.
  3. The Introduction section might be rewritten in another way. For instance: i) One paragraph could be devoted to mention the reasons that conductive polymer composites (CPCs) are novel materials with unique mechanical, electrical, thermal properties, etc. In this section/ paragraph, please mention some order of magnitude for these properties especially using TPU polymer matrix should be mentioned (i.e. the electrical and mechanical properties could be found elsewhere Materials 2020, 13(12), 2879) – ii) The second paragraph could be focused on graphene, methods for producing graphene and GO as well as reduced GO. In this paragraph, some polymer/rGO nanocomposites and their properties could be mentioned (RSC Adv., 2017, 7, 22145-22155) and iii) The third paragraph can be devoted to TPU/ nanofiller nanocomposites for mechanical and barrier properties; mentioning some existing literature, the state-of-the art values, etc. At the end of this paragraph the authors should mention what has been the trigger for their work and the main motivation not existing currently to the literature. This overall structure will make the Introduction part much more pedagogic and constructive for the reader.
  4. Line 85: the authors mention “Brabender Lab Station Torque Rheometer Plasti-Corder (Duisburg, Germany).” Is it possible to include some digital photo of this specific compounder, as well as give some more information about the screws, length of screw, etc. Please in the same paragraph mention also the sample geometry prepared by thermally assisted hot press.
  5. For the nanocomposites, could it be possible to include some TEM images as typically performed and reported for CPCs elsewhere (Polymer Volume 131, 22 November 2017, Pages 1-9) in order to show the “nano-scale” dispersion of the filler.
  6. In the XRD spectra, please include the Y axis intensity values, and the same also for the Raman spectra.
  7. For the mechanical properties section, do the authors have used any specific ASTM protocol?
  8. In the permeability section, it will be really nice to show with digital photos the different membranes that have been tested.
  9. Is it possible that the authors include some DSC results?

In terms of originality, importance & scientific quality, relevance & contribution to the field and presentation, this manuscript is of good level.

Furthermore, the discussion in different paragraphs could be improved in the points that have been indicated in order to make the manuscript more interesting to the reader and more educative.

The manuscript and its content are sufficiently novel to warrant its publication, however, after including and considering the additions and clarifications proposed.

Reviewer 2 Report

Maldonado-Magnere et al describe the influence of two thermally reduced graphene oxide (TRGO) containing 15.8% and 8.9% of oxygen as fillers on the barrier properties of TPU nanocomposites. They report a method for manufacturing composites from graphene oxide and reduced graphene oxide with high Young modulus. These properties are indeed difficult to combine if the graphene sheets have wrinkles. I recommend to accept this paper with moderate revisions.

  • Line 12 – ‘consist in’ should be ‘consist of’, please double check this.
  • Line 29 – filler should be written as fillers
  • Line 38-39, can authors please specify soft and hard segments clearly
  • Line 54 – which reducing agents are used to reduce graphene oxide into reduced graphene oxide. For example, hydrazine, sulfur etc.
  • Authors keep switching between TRGO and thermally reduced graphene oxide. Please make it consistent.
  • I suggest authors to refer graphene oxide synthesis to modified hummers method, which will make more sense to readers, given that authors have followed modified hummer methods, for examples authors can refer to the paper; https://doi.org/10.3390/cancers11030319
  • Line 70 – amount of graphene oxide used is missing, it is very important to initiate the reduction reaction, I therefore suggest authors to add the amount and concentration of graphene oxide used in reduction process.
  • Heading 2.4 – the samples preparation steps are missing for example how the sample was prepared for Raman spectroscopy?
  • Line 221 – the spelling of strength are incorrect.
  • The part to the high Young modulus (MPa), tensile strength (MPa) and elongation at break (%) in both composites should have high degree of organization from layer to layer. The authors achieve these properties using a simple manufacturing method. Can author explain how this technique is better that pressurised gyration or electrospinng. For example; https://doi.org/10.1016/j.compscitech.2020.108214. Such comparison can give a clear idea how this approach is better than existing approaches of manufacturing graphene-based composites.
  • Does oxygen content (or 600, 1000 temperature) has any effect on number of layers?
  • It seems to me that the improvement of organisation of composite assembly is only one part of achieving high mechanical features. Oxygen content coupled with temperature of making composites increases the thermodynamic minimum of binding the polymer to the inorganic sheets. In part they can see it on Raman scattering. However, the important thing would be to look at FTIR spectrum of the composite under different treatments. I am positive that the authors will see strong change in the bonding between the phases.
  • The mechanical properties obtained here are impressive and deserve due attention. However, I also think that the authors can make a step further and compare their results with other layered composites, single or few-layered graphene. SEM of graphene is also missing. The presence of wrinkles in reduced graphene is important for composites because of their higher rigidity, sharp edges and thus the properties can also rise. Could authors add some details on the morphology and architecture of reduced graphene oxide and how it influences the assembly of composites?
  • Although some of these studies deal with the composites of graphene that the authors are dealing with and with the same question how to organize them from the perspective of crystallinity and ‘vertical’ organization to attain high mechanical properties. Can authors discuss around this as well.
  • For reproducibility the average size of graphene oxide and reduced graphene oxide as well as their charge in the used dispersions need to be added to the characterization of the materials.
